# A Cross-Sectional, Multicentric, Disease-Specific, Health-Related Quality of Life Study in Greek Transfusion Dependent Thalassemia Patients

**DOI:** 10.3390/healthcare12050524

**Published:** 2024-02-22

**Authors:** Philippos Klonizakis, Noémi Roy, Ioanna Papatsouma, Maria Mainou, Ioanna Christodoulou, Despina Pantelidou, Smaro Kokkota, Michael Diamantidis, Alexandra Kourakli, Vasileios Lazaris, Dimitrios Andriopoulos, Apostolos Tsapas, Robert J. Klaassen, Efthymia Vlachaki

**Affiliations:** 1Adults Thalassemia Unit-2nd Department of Internal Medicine, Aristotle University, Hippokration General Hospital Thessaloniki, 54642 Thessaloniki, Greece; mfcmary@hotmail.com (M.M.); ioannchrist@auth.gr (I.C.); efivlachaki@yahoo.gr (E.V.); 2Department of Haematology, Oxford University Hospitals, NHS Foundation Trust, Oxford OX3 0AG, UK; noemi.roy@ouh.nhs.uk; 3Department of Mathematics, Imperial College London, London SW7 2BP, UK; i.papatsouma@imperial.ac.uk; 4Thalassemia Unit, AHEPA General Hospital of Thessaloniki, 54636 Thessaloniki, Greece; dpantelidou@yahoo.gr (D.P.); smaro_kokkota@hotmail.com (S.K.); 5Thalassemia and Sickle Cell Disease Unit, General Hospital of Larissa, 41110 Larissa, Greece; diamantidis76@gmail.com; 6Thalassemia and Hemoglobinopathies Center, University Hospital of Patras, 26504 Patras, Greece; akourakli@gmail.com (A.K.); vlazaris@gmail.com (V.L.); 7Haemato-Oncology Department, Royal Marsden Hospital NHS Foundation Trust, London SW3 6JJ, UK; dimitrios.andriopoulos@rmh.nhs.uk; 82nd Department of Internal Medicine, Aristotle University, Hippokration General Hospital of Thessaloniki, 54642 Thessaloniki, Greece; 9Department of Pediatrics, Division of Hematology/Oncology, University of Ottawa, Children’s Hospital of Eastern Ontario, Ottawa, ON K1H 8L1, Canada; rklaassen@cheo.on.ca

**Keywords:** Transfusion-Dependent Thalassemia, health-related quality of life, patient reported outcomes, disease-specific TranQol questionnaire, healthcare policy, Thalassemia infrastructure

## Abstract

The assessment of health-related quality of life (HRQoL) in thalassemia offers a holistic approach to the disease and facilitates better communication between physicians and patients. This study aimed to evaluate the HRQoL of transfusion-dependent thalassemia (TDT) patients in Greece. This was a multicentric, cross-sectional study conducted in 2017 involving 283 adult TDT patients. All participants completed a set of two QoL questionnaires, the generic SF-36v2 and the disease-specific TranQol. Demographic and clinical characteristics were used to predefine patient subgroups. Significant factors identified in the univariate analysis were entered into a multivariate analysis to assess their effect on HRQoL. The SF-36 scores of TDT patients were consistently lower compared to the general population in Greece. The mean summary score of TranQol was relatively high (71 ± 14%), exceeding levels observed in national surveys in other countries. Employment emerged as the most significant independent factor associated with better HRQoL, whereas age had the most significant negative effect. This study represents the first comprehensive QoL assessment of a representative sample of the TDT population in Greece. The implementation of TranQol allowed for the quantification of HRQoL in Greece, establishing a baseline for future follow-up, and identifying more vulnerable patient subgroups.

## 1. Introduction

Transfusion-dependent β-thalassemia (TDT) is a congenital disease characterized by reduced or no production of hemoglobin, leading to chronic anemia and necessitating lifelong blood transfusions [1]. During the last decades, TDT has gradually transformed from a life-threatening disorder into a chronic disease. The significant increase in survival rates has been attributed to the implementation of safe and systematic blood transfusions, the development of three effective iron chelators, and the guided follow-up of patients in well-organized thalassemia units. Nevertheless, the patients still confront the burden of the disease, which affects their functional status and health-related quality of life (HRQoL).

There is still an unmet need for a more holistic approach to assessing and managing health in thalassemia, aligning with the World Health Organization’s definition, which emphasizes that “health is a state of complete physical, mental, and social well-being and not merely the absence of disease”. The concept of HRQoL encompasses dimensions of life beyond traditional health indicators and can be delineated as “those aspects of self-perceived well-being that are related to or affected by the presence of disease or treatment” [2]. By measuring HRQoL in thalassemia, a comprehensive understanding of the disease burden emerges, facilitating improved communication among healthcare providers, patients, and their families [3]. Emerging research indicates that psychological well-being may serve as a protective factor against chronic physical ailments and promote longevity [4,5]. Consequently, the assessment of HRQoL has become a pivotal endpoint in the majority of clinical trials involving evolving treatments or interventions [6,7,8,9]. Especially in countries with a high prevalence of thalassemia, evaluating HRQoL status can inform and enhance social and healthcare policies, fostering better support for affected populations.

HRQoL can be assessed with quality of life (QoL) questionnaires completed by the patients themselves, constituting a prevalent form of patient reported outcome (PRO) [10]. Globally, very few reports have adequately addressed the HRQoL of adult thalassemia patients, with the majority utilizing the generic SF-36 questionnaire [11,12,13,14,15]. While generic QoL questionnaires facilitate comparisons with the general population, they may lack sensitivity for discerning QoL disparities among patient subgroups or evaluating intervention outcomes. In 2014, Klaassen et al. developed the TranQol, a disease-specific QoL questionnaire for TDT patients [16]. The conceptual framework for this new measure aims to comprehensively capture HRQoL dimensions in TDT patients, including social, emotional, and physical well-being domains [17]. The translated versions of TranQol, including the Greek adaptation, are actively utilized in international thalassemia clinical trials, such as the recently published phase 3 BELIEVE trial of Luspatercept [18]. However, real-world data regarding TranQol implementation beyond clinical trial settings remains limited.

Greece is among the countries with a high prevalence of TDT, with 19.4 cases per 100,000 individuals recorded from 2010 to 2015 [19]. However, there remains a dearth of evidence concerning disease-specific QoL among representative samples of the TDT patient population. In 2012, Lyrakos et al. developed and evaluated the psychometric properties of the Specific Thalassemia Quality of Life Instrument (STQOLI) to assess HRQoL in TDT patients [20]. Nevertheless, this instrument saw limited utilization in subsequent QoL studies, both within Greece and internationally. In all other studies involving Greek TDT patients, only generic questionnaires were employed among small patient cohorts, focusing on restricted domains of HRQoL, such as mental health and satisfaction with iron chelation therapy [12,21]. However, in 2017, the successful validation of the Greek version of TranQol offered the potential for a comprehensive and disease-specific assessment of HRQoL in TDT patients [22].

In this cross-sectional, multicentric study, our primary endpoint was to evaluate HRQoL in a representative cohort of Greek patients with TDT. This assessment involved employing both the generic SF-36v2 and the disease-specific TranQol questionnaire. The secondary endpoint was to investigate disparities in HRQoL among predefined patient subgroups, utilizing the high sensitivity of the disease-specific TranQol questionnaire. We hypothesized that various sociodemographic and clinical factors would exert an influence on the HRQoL of TDT patients in Greece. This research represents the inaugural examination of HRQoL, utilizing a combination of a generic and Thalassemia-specific questionnaire within a representative sample of TDT patients in Greece. The outcomes of this study are anticipated to establish a foundational benchmark for HRQoL, which can inform ongoing and forthcoming clinical trials involving TDT patients. Furthermore, our findings have the potential to highlight particularly vulnerable subgroups within the TDT patient population and contribute significantly to understanding the impact of the disease and available interventions on HRQoL.

## 2. Materials and Methods

### 2.1. Study Design and Setting

Τhis was a cross-sectional, multicentric study in a population of adult TDT patients that took place from October to December 2017. All patients were systematically transfused and closely monitored at four well-organized thalassemia units in Greece. The study sites were all located in urban areas of Greece and involved the adult Thalassemia Unit of the Second Department of Internal Medicine, Aristotle University, Hippokration General Hospital of Thessaloniki, the Thalassemia Unit of the AHEPA General Hospital of Thessaloniki, the Thalassemia and Sickle Cell Disease Unit of the General Hospital of Larissa, and the Thalassemia Unit of the University Hospital of Patras. We evaluated HRQoL using a comprehensive approach that combined two distinct quality-of-life assessment tools. Firstly, we utilized the generic Greek SF-36v2 [23], which measures both physical and mental QoL domains. This allowed for comparisons of HRQoL between our study TDT patients and the general population. Secondly, we employed the validated Greek version of the disease-specific TranQol questionnaire [22]. This instrument is more sensitive in revealing differences among patient subgroups and offers a detailed evaluation of overall HRQoL and its five subdomains: physical health (PH), emotional health (EH), sexual health (SH), family functioning (FF), and school and career functioning (SCF).

### 2.2. Inclusion and Exclusion Criterion

The participants were selected according to predefined and specific inclusion and exclusion criteria. Briefly, the participants were aged 18 years and older, had a confirmed diagnosis of transfusion-dependent beta-thalassemia, and had efficient reading and writing skills without significant intellectual or sensory disorders. We excluded patients who were participating in a clinical trial, were pregnant, or had been recently hospitalized on the grounds of avoiding factors that could cause any transient change in the participants’ QoL.

### 2.3. Sampling and Sample Size

To reflect real life and avoid selection bias, all TDT patients from each thalassemia unit were invited consecutively. During their routine transfusion visit, we asked the participants to complete the two quality-of-life questionnaires, SF-36v2 and TranQol. In terms of the study’s secondary endpoint, we prespecified patient subgroups that might have different QoL according to clinical, demographic, and socioeconomic status. Thus, we looked for differences between age groups, gender, family structures, educational backgrounds, working status, comorbidities, types of iron chelation therapy, and hemosiderosis status. We used the MRI T_2_* examination, a standard of care for all the participating thalassemia units, to assess iron overload status and define patients with no heart or no hepatic iron overload (heart T_2_* > 20 ms and liver iron concentration (LIC) ≤ 3 mg Fe/g dry weight, respectively) [1,24]. All data were retrieved from the patients’ medical records, and the variables collected are listed in Table 1.

We determined the study population sample size according to the study’s primary aim, which was to evaluate HRQoL in the Greek TDT patient population. A representative sample was estimated to be 242 patients for the total 2099 TDT population in Greece [19], with a 90% confidence interval and a 5% margin of error [25]. The total study population was set to be at least 403 patients, taking into account a 40% nonresponse rate and a minimum 60% response rate (RR), respectively (definition proposed by the American Association of Public Opinion Research, AAPOR RR6) [26,27]. We set the power to 0.8 (80%) and a 5% significance level to have a high probability of detecting a minimal clinically important difference (MCID) between patient subgroups in terms of the secondary endpoint of the study. The minimal sample size was estimated to be 64 patients for each subgroup [25]. A minimal clinically important difference (MCID) in SF-36v2 and TranQol scores was defined according to Norman et al. [28], who suggested that the value of half a standard deviation (0.5 × SD) of baseline scores can serve as a default value for important patient-perceived change on QoL measures used with patients with chronic diseases. In this study, the MCID was calculated from the half a standard deviation (0.5 × SD) of previously published SF-36v2 and TranQol scores, in a survey of 94 Greek TDT patients (Table 2) [22].

### 2.4. Statistical Tools

We analyzed qualitative and quantitative variables descriptively, with qualitative data presented as frequencies and quantitative data as mean ± standard deviation (SD). We calculated TranQol scores from a mathematical equation provided by the developers [16] and the SF-36 scores from the QualityMetric Health Outcomes^TM^ Scoring Software 4.5 [29]. Higher scores on both questionnaires indicate better QoL (0 indicates the worst possible health state, and 100 indicates the best health state). The TranQol and SF-36 were considered complete if ≥75% and ≥50% of the questionnaire items were answered, respectively [16,30] We used the “½ SD method”, a distribution-based approach that has been developed to define clinically meaningful differences in each QoL questionnaire score [28]. Any differences of 0.5 × SD on previously published TranQol and SF-36v2 scores were considered clinically significant [22]. We performed univariate linear regression analysis to assess the association between independent variables (age, gender, working status, educational status, marital status, offspring, heart and liver MRI T_2_* status, co-morbidities, and type of chelation therapy) and TranQol summary and subdomain scores. We used a less-restrictive α-level of 0.2 in univariate linear regression analysis to identify a broad range of factors that would be retained as candidate variables in the next step of multivariate analysis. We conducted stepwise regression to select the most parsimonious models of independent explanatory factors that may be associated with overall and subdomain QoL status, after adjusting for confounding factors. We employed a less restrictive alpha level of 0.1 to identify a broader range of independent factors, indicating statistical trends for values where 0.05 < *p* < 0.10. The multivariate analysis data were checked and met the assumptions of homogeneity of variance and linearity, and the residuals were approximately normally distributed. For each of the significant predictors, we estimated the regression coefficients (B) and their 95% confidence intervals. The data were analyzed using IBM SPSS version 27.0 (IBM Corp., Armonk, NY, USA).

### 2.5. Ethical Considerations

The study was approved by the Ethics Committee, and written informed consent was obtained from each patient.

## 3. Results

### 3.1. Sociodemographic and Clinical Characteristics of Study Participants

A total of 474 adult TDT patients were screened from four thalassemia units, of which 286 (60%) met eligibility criteria and consented to participate in the study. The primary reason for exclusion was patient refusal, accounting for 22% of those screened. Additionally, 10% of patients were excluded for not meeting the study’s inclusion/exclusion criteria, whereas no reason was recorded for the remaining 8% of patients who were not recruited. The response rates of the patients who completed the SF-36v2 (≥50% items) and the TranQol (≥75% items) questionnaires were 61% (*n* = 227) and 66% (*n* = 283), respectively (Figure 1 flow diagram). The sociodemographic and clinical characteristics of the study population are summarized in Table 1. The participants’ mean ± SD age was 39 ± 9 years (range:18–68), and 58% were female. Almost half of the study population were married (46%) and had offspring (42%). Most participants had a higher educational level (77%), and 59% were employed. Among the participants with a history of co-morbidity (68%), the most frequent disorder was osteoporosis (39%). Regarding chelation therapy, Deferasirox monotherapy (35%) or the combination of Deferoxamine/Deferiprone (39%) were the two most frequently used iron chelating regimens. Only 7% of the participants had an abnormal heart MRI T_2_*, whereas 39% of the participants had an abnormal liver MRI T_2_*.

The Response Rate 6 definition proposed by the American Association of Public Opinion Research (AAPOR) was used to calculate response rates.

### 3.2. Health-Related Quality of Life in Transfusion-Dependent Thalassemia Patients

The TranQol and SF-36 summary and subdomain scores in the study population are reported in Table 2. The overall mean ± SD TranQol summary score was 71 ± 14%, and the overall mean ± SD of SF-36v2 PCS and MCS scores were 51 ± 8% and 49 ± 9%, respectively. In the same table, we report the SF-36v2 and TranQol MCID values that represent a threshold for meaningful changes in each questionnaire’s score. In Figure 2, we present the comparison of HRQoL between TDT patients and the general population. The SF-36 subdomain scores from our survey of TDT patients were substantially lower compared to the only available data on SF-36 scores in the general Greek population, published in 2005 [23], and the differences were clinically important (>MCID). On the other hand, the TranQol scores from our survey were higher compared to national surveys in the United Kingdom [31], Canada [16], and Peshawar, Pakistan [32], and the differences in TranQol summary and most subdomain scores were clinically important (Figure 3).

### 3.3. Factors That Influence Health-Related Quality of Life among TDT Patients

In terms of statistical trends, the factors that were associated with summary TranQol scores in univariate regression analysis included age (B = −0.3, 95% CI −0.5,−0.2), marriage (B = −3.2, 95% CI −6.5,0.2), employment (B = 7.5, 95% CI 4.2,10.7), higher education (B = 4.2, 95% CI 0.4,7.9), comorbidities (B = −5.1, 95% CI −8.6,1.6), iron chelation with Deferasirox compared to Deferoxamine monotherapy (B = 4.1, 95% CI −0.4,8.5), and abnormal liver MRI T2* (B = 3.2, 95% CI −0.3,6.7) (Table 3). Abnormal heart MRI T2* (B = −2.6, 95% CI −8.4,3.3) was negatively associated only with the physical subdomain. Female gender exhibited a significant association with higher scores in family functioning (B = 4.5, 95% CI 0.2,8.6) and school and career (B = 7.2, 95% CI 1.9,12.5) subdomains. In multivariate regression analysis (Table 4), age, female gender, employment, and iron chelation with Deferasirox compared to Deferoxamine monotherapy were the independent factors that significantly influenced HRQoL, after adjusting for confounding factors. Based on the multivariate model, there was on average a decrease of 0.3 units in TranQol summary scores for every 1-year increase in age (B = −0.3, 95%CI 0.5,−0.02), corresponding to a 20-year increase in age for a TranQol MCID of 6 units (clinically important increase in age = 6/0.3). Employment was associated with a significant average increase in TranQol summary, physical, emotional, and family functioning scores, and the levels of increase were greater than the MCID reported in Table 2, reflecting clinically meaningful changes. Female gender was associated with a significant increase in the TranQol family functioning (B = 7.4, 95% 1.5,13.4), and school and career (B = 10.8, 95% 3.3,18.2) subdomain scores, and the change was clinically important (B > MCID, Table 2). Deferasirox compared to Deferoxamine monotherapy was only associated with an average increase in the TranQol physical subdomain score (B = 5.8, 95%CI 0.01,11.6), and the change was marginally clinically important (MCID = 6, Table 2).

## 4. Discussion

This survey marks a significant milestone as the first comprehensive assessment of QoL in a representative sample of the adult TDT population in Greece. Employing a combination of the generic SF-36v2 and the thalassemia-specific TranQol questionnaires, our study sheds light on the nuanced aspects of HRQoL in this patient cohort. Our findings reveal a concerning trend: adult TDT patients exhibit significantly lower HRQoL compared to the general Greek population, with notable differences observed in both physical and mental health domains. Despite considerable advancements in the clinical management and treatment of TDT patients, our results underscore a critical need for further optimization of QoL (Figure 2). Our findings align with existing literature, which reports a significant negative impact of TDT on the physical and mental health status of adult patients [11,13,32,33,34,35]. Similarly, studies focusing on pediatric TDT patients generally agree that HRQoL falls below population norms, with a more pronounced difference observed in adolescents compared to children [36,37,38,39]

Intriguingly, our study also reveals relatively high TranQol scores in Greece, indicative of a comparatively good level of HRQoL among TDT patients (Figure 3). Specifically, the TranQol summary score in our survey was 73%, a clinically important higher score compared to the TranQol summary score of 54% in a recently published Global Longitudinal Study of HRQoL in adult TDT patients (preliminary results) in the U.S.A., United Kingdom, France, Italy, and Germany [40]. Moreover, while the Summary and TranQol subdomain scores in our study were marginally higher compared to the TranQol results from the phase 3 BELIEVE trial of luspatercept across 15 countries (Australia and countries across Europe, the Middle East, North Africa, North America, and Southeast Asia), these differences were not clinically important [18]. The relatively high TranQol scores in Greece may be attributed to the established special infrastructure, including well-organized medical centers, access to medical resources, and the availability of blood units and iron chelators, coupled with social, educational, and professional benefits. It is important to note that such a supportive social and healthcare environment may not be available in all countries, as healthcare policies and services for TDT patients vary considerably [41]. The notable disparity in HRQoL between Greece and Pakistan (TranQol Summary: 73% vs. 43%, Figure 3) addresses the need to promote health equity in thalassemia across the world, especially in light of emerging novel drugs and therapeutic strategies.

After controlling for possible confounding sociodemographic and clinical variables, as demonstrated in the multivariate analysis presented in Table 3, we identified older age and unemployment as the two most significant factors associated with lower HRQoL. The absence of higher education, comorbidities, and iron chelation with Deferoxamine monotherapy were additional factors that negatively affected certain subdomains of HRQoL.

Our results are in alignment with previous studies reporting that age is independently associated with reduced QoL in TDT [14,15,31,42]. However, it is noteworthy that most researchers agree that in the general population, older adults exhibit social and emotional functioning that is equal to or superior to that of younger adults [4,43]. Sobota et al. compared the HRQoL in TDT with the US norms and showed that HRQoL is lower in older TDT patients than would be expected in the general population [15]. The impact of age on HRQoL in TDT patients should be interpreted cautiously, considering that older TDT patients may have grown up in less favourable healthcare environments, lacking access to oral iron chelators and the improved thalassemia infrastructure available today. Conversely, TDT, being a chronic disease, may have a cumulative negative effect on HRQoL, akin to other individuals with disabilities [44]. Further investigation, particularly through close follow-up of younger TDT patients, may provide insights into the effect of age on HRQoL.

Employed TDT patients exhibited both significantly and clinically higher TranQol summary and subdomain scores (physical health, emotional health, and family functioning). Existing literature suggests a positive relationship between employment and HRQoL among persons with disabilities [44]. However, there is limited published data on the effect of employment on HRQoL in TDT. Most studies have utilized generic QoL questionnaires, which may not be as sensitive as TranQol in detecting the impact of employment on HRQoL. A recent survey in Saudi Arabia was among the few studies reporting that TDT participants holding professional jobs had better mental HRQoL scores compared to those engaged in clerical or manual occupations [45]. Similarly, another survey in Bangladesh demonstrated that lower income was associated with a deterioration of HRQoL [46]. It is important to note that employment rates for TDT patients vary across countries. In our study population, 59% of patients were employed, while a study in the Middle East reported an employment rate of 54% [47]. Conversely, preliminary data from the Global Survey conducted in the U.S.A., United Kingdom, France, Italy, and Germany indicated that only 34% of TDT patients were employed [40]. Our findings underscore the significance of established employment policies for TDT patients in Greece, which provide job opportunities in the public sector and the potential for earlier retirement with a fully paid pension. Nevertheless, creating an accessible and suitable working environment for TDT patients remains a challenge for policymakers, especially considering recent findings by Shah et al., which indicated that the mean working productivity impairment for TDT patients was 42% compared to a typical working week [31].

Gender and choice of iron chelator were significantly associated only with certain domains of HRQoL. As shown in Table 4, female patients exhibited significantly higher TranQol scores in the family functioning and school and career subdomains. There is no consensus in the literature regarding the effect of gender on HRQoL in TDT. Sobota et al. reported that female patients in the USA had worse HRQoL [15], whereas studies in Iran and Bangladesh demonstrated that female patients had a better quality of life [39,46,48]. It could be speculated that the differences in cultural and social environments between countries may explain the contradictory effect of gender on HRQoL.

Regarding comparisons between different iron chelators, Deferasirox was positively associated only with the physical health subdomain compared with Deferoxamine monotherapy, with no difference in HRQoL compared to patients treated with Deferoxamine combined with oral Deferiprone. The published data regarding the effect of iron chelators on HRQoL are heterogeneous in the literature. Sobota et al., in one of the largest studies of TDT patients, concluded that the type of chelator was not associated with HRQoL and suggested that this may be due to patients typically being free to choose their chelation therapy [15]. Goulas et al., in a study of quality of life and iron chelation treatment satisfaction, showed that TDT patients receiving Deferoxamine, alone or in combination with an oral iron chelator, perceived higher treatment efficacy, comparable HRQoL, and comparable satisfaction with the iron chelator compared to patients receiving Deferoxamine, alone or in combination with an oral iron chelator [12]. In clinical practice, physicians may utilize our findings to inform patients about the choice of iron chelator and address any concerns about their impact on everyday life.

This study had some limitations. The cross-sectional design of the study may be less informative in detecting differences in HRQoL between patient subgroups compared to a longitudinal prospective design. Additionally, all participants were recruited from reference thalassemia centers in urban areas of Greece, so our results may not fully reflect HRQoL among TDT patients residing in rural areas and treated in less organized thalassemia institutions. Nevertheless, the study had a high participation rate and a high probability of detecting significant differences between almost all patient subgroups (power = 0.8, significance level = 5%), representing real-world clinical practice in a routine care environment.

The assessment of HRQoL status among TDT patients in Greece has been an unmet need until now. In this multicentric study, we have successfully measured HRQoL using the combination of the generic SF-36v2 and the disease-specific TranQol questionnaires, with an adequate response rate from the participants in a representative sample of the Greek TDT population. We have demonstrated that both social-demographic and clinical factors may impact the HRQoL of TDT patients. The implementation of the disease-specific TranQol offers the potential for future re-evaluation of HRQoL and the identification of emerging factors affecting HRQoL in the era of novel treatments and interventions in TDT.

## 5. Conclusions

The integration of TranQol, an internationally recognized disease-specific QoL questionnaire, has provided us with a robust framework to assess the HRQoL of TDT patients in Greece. This approach not only enabled us to quantify HRQoL levels but also established a baseline for future monitoring and comparison with other countries. The identification of vulnerable patient subgroups, particularly older individuals and those who were unemployed, underscores the need for targeted interventions to enhance their QoL. Overall, the HRQoL status in TDT is relatively high in Greece, possibly due to the healthcare environment and social support. Other countries with high incidence rates of TDT may adopt the “paradigm of Greece” and improve the HRQoL of TDT patients.

## Figures and Tables

**Figure 1 healthcare-12-00524-f001:**
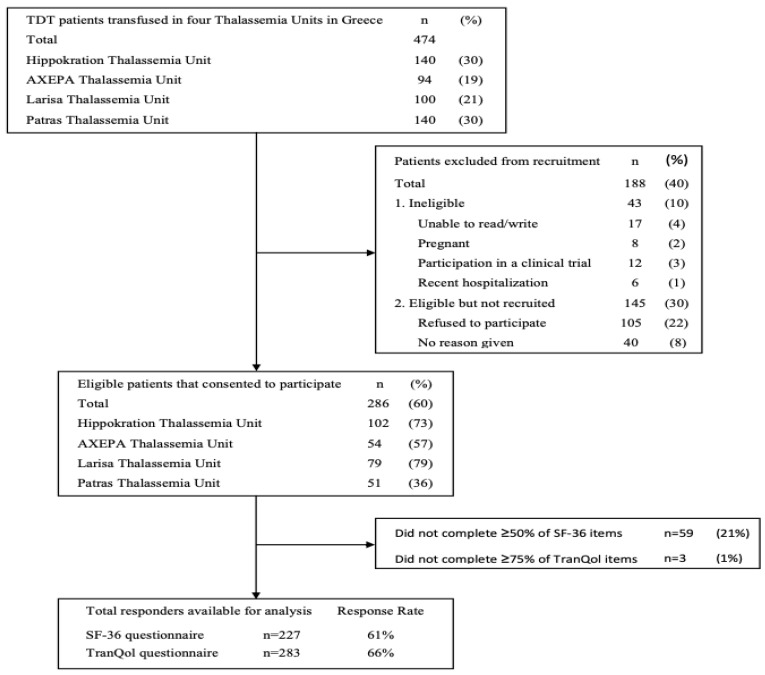
Flow diagram of participants and response rate.

**Figure 2 healthcare-12-00524-f002:**
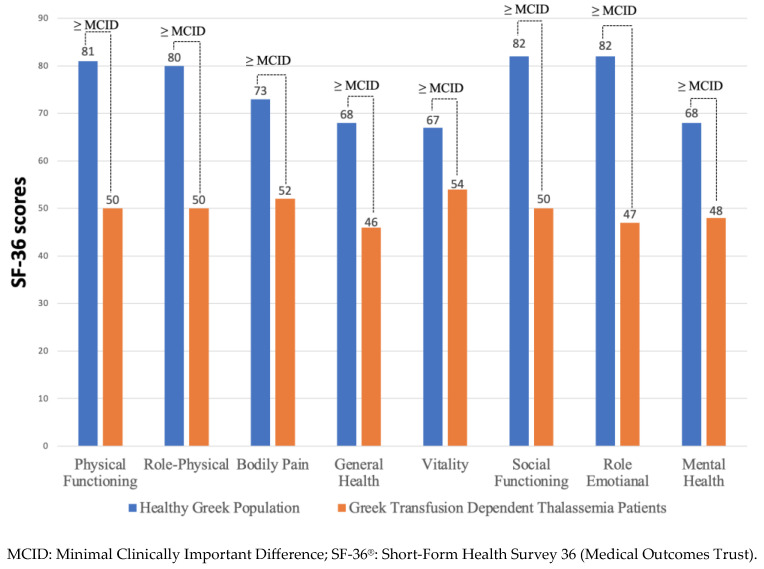
Comparison of the SF-36 scores between the present study and the study in the general Greek population provided by Pappa et al. The bar charts depict the differences in the mean scores between the two studies. The minimal clinical important difference (MCID), defined as the 0.5 × SD on previously reported SF-36v2 scores in TDT patients, was used as a threshold value to demonstrate clinically meaningful differences in QoL between the general population and TDT patients.

**Figure 3 healthcare-12-00524-f003:**
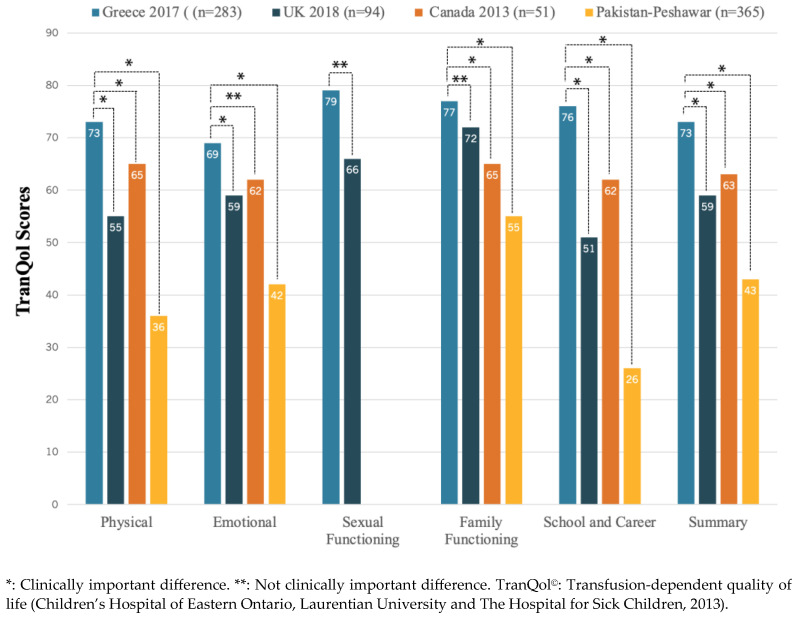
The TranQol summary and subdomain scores from national surveys conducted in different countries are compared and visually represented using different colored bars. We used the minimal clini-cal important difference (MCID), defined as the 0.5 × SD on previously reported TranQol scores in TDT pa-tients, as a threshold value to demonstrate clinically meaningful differences in HRQoL between national surveys in different countries.

**Table 1 healthcare-12-00524-t001:** Sociodemographic and clinical characteristics of the study population.

Characteristics	Parameter Value
Sex, N (%)	
Male	110 (39)
Female	165 (58)
NA	11 (4)
Age, mean ± SD (range) years	39 ± 9 (18–68)
Age Groups N (%)	
Under 25 yrs	20 (8)
26–35 yrs	70 (27)
36–45 yrs	115 (44)
46–55 yrs	41 (16)
Above 56	14 (5)
NA	26 (9)
Marital status	
Single	143 (54)
Married	124 (46)
NA	19 (7)
Offsprings, N (%)	
Yes	114 (42)
No	156 (58)
NA	15 (5)
Educational Level, N (%)	
Basic	38 (23)
Higher	127 (77)
NA	121 (42)
Employment Status, N (%)	
Unemployed	110 (41)
Employed	158 (59)
NA	18 (6)
Co-Morbidities, Yes N (%)	
No	89 (31)
Yes	191 (68)
Splenectomy	82 (29)
Osteoporosis	112 (39)
Hypothyroidism	74 (26)
Hypogonadism	76 (27)
Diabetes	19 (7)
NA	6 (2)
Type of Chelator N (%)	
Deferasirox	90 (35)
Deferoxamine	52 (20)
Deferiprone	15 (6)
Deferoxamine/Deferiprone	99 (39)
Abnormal Liver MRI T_2_ * N (%)	101 (39)
Abnormal Heart MRI T_2_ * N (%)	20 (7)

Abbreviations: MRI T_2_*; T_2_ Star Magnetic Resonance Imaging pulse sequence.

**Table 2 healthcare-12-00524-t002:** TranQol and SF-36v2 scores in transfusion-dependent thalassemia patients and Minimal Clinically Important Difference values.

Quality of Life Questionnaires		Scores% (Mean ± SD)	MCID (0.5 × SD *) ^2^
SF-36v2 ^1^	Physical Functioning	50 ± 7	≥8
	Role-Physical	50 ± 9	≥11
	Bodily Pain	52 ± 10	≥10
	General Health	46 ± 10	≥12
	Vitality	54 ± 9	≥11
	Social Functioning	50 ± 8	≥9
	Role Emotional	47 ± 10	≥11
	Mental Health	48 ± 9	≥9
	Physical Component Summary	51 ± 8	≥5
	Mental Component Summary	49 ± 9	≥5
TranQol ^1^	Physical Health	71 ± 15	≥7
	Emotional Health	68 ± 17	≥8
	Sexual Health	78 ± 21	≥16
	Family Functioning	75 ± 17	≥10
	School and Career	76 ± 21	≥12
	Summary	71 ± 14	≥6

The table reports the TranQol and SF-36v2 summary and subdomain scores in the study population and the corresponding minimal clinically important difference values. TranQol^©^: Transfusion-dependent quality of life (Children’s Hospital of Eastern Ontario, Laurentian University, and The Hospital for Sick Children, 2013); SF-36^®^: Short-Form Health Survey 36 version 2 (Medical Outcomes Trust). ^1^ The TranQol and SF-36 questionnaires are scored on a 0 (poor) to 100 (good health) scale. MCID: Minimal clinically important difference, defined according to Norman et al. [28], who suggested that the value of half a standard deviation (0.5 × SD) of baseline scores can serve as a default value for important patient-perceived change on QoL measures used with patients with chronic diseases. ^2^ (0.5 × SD *): calculated 0.5 × SD on previously published TranQol and SF-36v2 scores in TDT patients.

**Table 3 healthcare-12-00524-t003:** Univariate analysis of covariates associated with TranQol Summary and subdomain scores.

* TranQol Summary and Subdomains																		
	Summary			Physical Health			Emotional Health			Sexual Health			Family Functioning			School and Career		
Variables	B	S.E	95%CI	B	S.E	95%CI	B	S.E	95%CI	B	S.E	95%CI	B	S.E	95%CI	B	S.E	95%CI
**Age** (**yrs**)	**−0.3 ***	0.1	(−0.5, −0.2)	**−0.4 ***	0.1	(−0.6, −0.2)	**−0.3 ***	0.1	(−0.5, −0.03)	**−0.3 ***	0.2	(−0.7, −0.01)	**−0.5 ***	0.1	(−0.7, −0.3)	−0.1	0.2	(−0.5, 0.2)
**Marital Status** Married Ref:Unmarried	−3.2 **	1.7	(−6.5, 0.2)	−3.2 **	1.8	(−6.8, 0.4)	−3.8 **	2.0	(−7.8, 0.1)	0.5	2.8	(−4.9, 5.9)	−1.8	2.1	(−5.9, 2.3)	−0.1	2.7	(−5.4, 5.3)
**Employment** Yes Ref:No	**7.5 ***	1.7	(4.2, 10.7)	**6.6 ***	1.8	(3.0, 10.2)	**6.4 ***	2.0	(3.0, 10.2)	**6.4 ***	2.8	(0.8, 12.0)	**8.6 ***	2.1	(4.5, 12.7)	**8.3 ***	2.8	(2.9, 13.8)
**Educational Status** Higher Ref:Not Higher	**4.2 ***	1.9	(0.4, 7.9)	**5.5 ***	2.1	(1.5, 9.5)	1.5	2.3	(−3.0, 5.9)	−5.6 **	3.3	(−11.9, 0.8)	4.5 **	2.4	(−0.2, 9.1)	**7.5 ***	3.2	(1.3, 13.7)
**Comorbidities** Yes Ref:No	**−5.1 ***	1.8	(−8.6, 1.6)	−3.7 **	1.9	(−7.5, 0.2)	**−4.9 ***	2.1	(−9.1, 0.8)	**−8.8 ***	2.9	(−14.5, −3.1)	**−8.2 ***	2.2	(−12.5, −3.8)	−3.1	2.9	(−8.7, 2.6)
**Iron Chelator**																		
DFX Ref:DFO	4.1 **	2.2	(−0.4, 8,5)	**6.9 ***	2.5	(2.0, 11.9)	3.4 **	2.6	(2.0, 11.9)	6.6 **	3.9	(−1.2, 14.3)	4.3 **	3.1	(−1.7, 10.3)	−6.1 **	3.6	(−13.3, 1.0)
DFX Ref:DFO + DFP	0.9	2.0	(−3.0, 4.9)	3.0 **	2.1	(−1.2, 7.2)	1.4	2.4	(−1.2, 7.2)	3.3	3.3	(−3.2, 9.8)	−2.5	2.5	(−7.3, 2.4)	−4.8 **	3.1	(−11.0, 1.3)
**Liver MRI T2** * Abnormall Ref:Normal	3.2 **	1.8	(−0.3, 6.7)	2.9 **	1.9	(−0.9, 6.7)	3.9 **	2.1	(−0.2, 8.0)	1.3	2.9	(−4.5, 7.0)	1.5	2.2	(−2.8, 5.8)	−0.1	2.8	(−5.5, 5.4)
**Heart MRI T2** * Abnormal Ref:Normal	−2.6	3.0	(−8.4, 3.3)	−3.9 **	3.2	(−10.2, 2.5)	−1.7	3.5	(−8.7, 5.2)	−2.4	4.9	(−12.1, 7.3)	−0.9	3.7	(−8.1, 6.4)	−0.9	4.8	(−10.5, 8.5)
**Gender** Female Ref:Male	1.6	1.7	(−1.8, 5.0)	−0.6	1.4	(4.1, 3.3)	−0.4	1.5	(−4.3, 3.8)	1.9	2..1	(−2.1, 8.9)	**4.4 ***	2.1	(0.2, 8.6)	**7.2 ***	2.7	(1.9, 12.5)
**Offsprings** Yes Ref:No	−1.7	1.6	(−4.8, 1.3)	−2.9 **	1.7	(−6.2, 0.4)	−1.3	1.9	(−4.9, 2.4)	−0.1	2.8	(−5.3, 5.6)	−0.9	1.9	(−4.8, 2.8)	0.9	2.7	(−4.5, 6.2)

A negative correlation coefficient indicates that a patient subgroup has worse QoL compared to the reference subgroup. Highlighted effects are statistically significant. A less restrictive *a*-level of 0.2 was used to identify factors that would be retained as candidate variables in multivariate analysis. Ref: Reference; *: *p*-value < 0.05; **: *p*-value < 0.2; B: Standardized beta coefficient; CI: Confidence interval; DFX: Deferasirox; DFO: Deferoxamine; DFP: Deferiprone.

**Table 4 healthcare-12-00524-t004:** Multivariate model of significant factors (*p* < 0.05) associated with HRQoL.

TranQol	Age			Female Gender			Employed			DFX (Ref:DFO)		
	B	S.E	95%CI	B	S.E	95%CI	B	S.E	95%CI	B	S.E	95%CI
**Summary**	−0.3	0.1	(−0.5, −0.02)	-		-	**7.5**	2.4	(2.8, 12.2)	-		-
**Physical Health**	−0.3	0.1	(−0.6, −0.01)	-		-	**5.8**	2.8	(0.2, 11.4)	5.8	2.9	(0.01, 11.6)
**Emotional Health**	−0.3	0.1	(−0.6, −0.01)	-		-	**8.1**	2.8	(2.5, 13.6)			
**Sexual Health**	-	-	-	-		-	**-**		-	-		-
**Family Functioning**	−0.6	0.2	(−0.9, −0.3)	**7.4**	3.0	(1.5,13.4)	**8.2**	3.0	(2.2, 14.2)	-		-
**School and Career**	-	-		**10.8**	3.8	(3.3,18.2)	-		-	-		-

Stepwise linear regression was conducted to select the most parsimonious models of independent explanatory factors that may be associated with overall and subdomain QoL status, after adjusting for confounding factors. A less-restrictive a-level of 0.1, in terms of statistical trends (0.05 < *p* <0.10), was used to identify a broader range of independent factors. The multivariate analysis data were checked and met the assumptions of homogeneity of variance and linearity, and the residuals were approximately normally distributed. For each of the significant predictors, we estimated the regression coefficients (B), standard error and the 95% confidence intervals. Highlighted Beta coefficients imply a clinically meaningful change in TranQol score according to predefined Minimal Clinically Important Difference. B: standardized beta coefficient; S.E: Standard Error; CI: Confidence Interval; Ref: Reference; DFX: Deferasirox; DFO: Deferoxamine.

## Data Availability

The authors confirm that the data supporting the findings of this study are available within the article.

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
