# Peer review of "A Cross-Sectional, Multicentric, Disease-Specific, Health-Related Quality of Life Study in Greek Transfusion Dependent Thalassemia Patients"

_healthcare, 2024, doi:10.3390/healthcare12050524_

Round 1

Reviewer 1 Report

Comments and Suggestions for Authors

I find this a very interesting and well presented study. It shows clearly the difference between generic and disease specific tools in assessing the quality of life. As then authors say in discussion, the patient population was derived from refence centres and the result may be different if patients from peripheral centres are assessed (? the Greek islands). I urge the researchers to continue with such an assessment as this may also help in comparing patients under very different quality of care conditions, such as between reference centres and patients treated in LMIC counties.     

Author Response

Thank you for your positive feedback and insightful comments on our article titled "A Cross-Sectional, Multicenter, Disease-Specific, Health-Related Quality of Life Study in Greek Transfusion Dependent Thalassemia Patients" We appreciate your careful review and value your suggestions. We are indeed planning to assess HRQoL in Thalassemia patients living in rural areas and treated in non-reference Thalassemia centers. Our study results could be used as a comparator, to identify any differences in HRQoL between patients that are followed up in reference Thalassemia centers and those that do not have full access to the required healthcare services.

Reviewer 2 Report

Comments and Suggestions for Authors

HRQoL for patients with any chronic illness like thalessemia is extremely important as it impacts QOL since childhood. This study aims to capture factors impacting QOL in adult patients with thalessemia  requiring transfusion. 

Few minor edits:

Introduction: Would be important to highlight prevalence of thalassemia in Greece. Though Pediatric QOL has been published there is scarce literature on adult patients with thalessemia in mediterranean countries. 

1. Page:1 Line :44 TDT is a congenital(-) characterized- Please complete the sentence with congenital disease/disorder. 

Page: 2 Line 96: In methodology- Please elaborate on secondary end point. What was the prespecified subgroup? Also clarify if only beta thal included or any other variant. 

Did all patients get MRI liver and heart? If that is SOC at institutes please mention that or was it based on  Labs(ferritin levels, transaminitis ) or clinical concern. 

Discussion: Would be important to understand differences between pediatric and adult HRQOL in thalessemic patients. Would be more impactful if 1-2 lines are mentioned on this comparing to published literature for pediatric population. 

Author Response

We appreciate your review of our paper " A Cross-Sectional, Multicenter, Disease-Specific, Health-Related Quality of Life study in Greek Transfusion Dependent Thalassemia Patients"

Your feedback is valuable, and we addressed the issues you raised to enhance the quality and impact of our article.

  1. Introduction: Would be important to highlight prevalence of thalassemia in Greece. Though Pediatric QOL has been published there is scarce literature on adult patients with thalessemia in mediterranean countries. 

We understand the importance of presenting the prevalence of thalassemia in Greece. We have added the following sentence in the introduction and the relevant reference (first line of the 4th paragraph, Lines 84-85) :“Greece is among the countries with a high prevalence of TDT, with 19.4 cases per 100,000 individuals during the period from 2010 to 2015[19]”

  1. Page:1 Line :44 TDT is a congenital(-) characterized- Please complete the sentence with congenital disease/disorder. 

We apologize for the typo error. We have corrected the sentence using the phrase congenital disease.

  1. Page: 2 Line 96: In methodology- Please elaborate on secondary end point. What was the prespecified subgroup? Also clarify if only beta thal included or any other variant. 

Thank you for your comments.

We have better described the study’s secondary endpoint in the methods, as follows (lines 142-147): “In terms of the study’s secondary endpoint, we prespecified patient subgroups that might have different QoL according to clinical, demographic and socioeconomic status. Thus, we looked for differences between age groups, gender, family structures, educational backgrounds, working status, comorbidities, types of iron chelation therapy and hemosiderosis status.”

 Only Transfusion Dependent beta-Thalassemia patients were included in the study. We have clarified this in the inclusion criteria as follows (lines 133-134): Briefly, the participants were aged 18 years and older, had a confirmed diagnosis of transfusion-dependent beta-Thalassemia.

  1. Did all patients get MRI liver and heart? If that is SOC at institutes please mention that or was it based on  Labs(ferritin levels, transaminitis ) or clinical concern. 

We acknowledge your point about the iron overload estimation methods. MRI T2* is the standard of care in reference Thalassemia centers in Greece. We have added the following comment in the methods section (lines 147-149): “We used the MRI T2* examination, a standard of care for all the participating Thalassemia Units, to assess iron overload status and define patients with no heart or no hepatic iron overload”

  1. Discussion: Would be important to understand differences between pediatric and adult HRQOL in thalessemic patients. Would be more impactful if 1-2 lines are mentioned on this comparing to published literature for pediatric population. 

We understand your concern about the importance of understanding differences between pediatric and adult HRQoL in Thalassemic patients. In response to your suggestion, we have added the following sentence and the relevant reference (Discussion, lines 379-381): : Similarly, studies focusing on pediatric TDT patients generally agree that HRQoL falls below population norms, with a more pronounced difference observed in adolescents compared to children[36-39]”

Reviewer 3 Report

Comments and Suggestions for Authors

Thank you for allowing me the opportunity to review the article. The authors have done a multicentric Cross-Sectional study assessing the Quality of Life study in Greek Transfusion Dependent 3 Thalassemia Patients. I appreciate the authors for their hard work, but there are certain concerns which need to be addressed :

1.   Novelty and relevance: The article adds important data and is relevant but the authors must enhance the last paragraph of introduction even more - and underline the relevance of study in detail. Make the objectives and endpoints more in line with your title to allow better understanding for the readers. It will also be wise to suggest in a single line how this study is different from already existing body of literature.

2.     Academic writing, grammar, flow and readability: Overall writing is good. But use of word "multicentre" can better be replaced with "multicentric". Also there are various areas where passive voice has been extensively used. It will be better to report in active voice as that helps better engagement for the reader.

3.     Narrative structure: Great structure and headings.

4.     Methodology: This section can be arranged better - the paragraph though descriptive is kind of confusing for the reader. It can better be placed in followinbg headings: Study design and setting, Inclusion and Exclusion criterion, Sampling and Sample Size, Statistical tools and Ethical considerations.

5.     Statistical tools employed: Good description.

6.     Discussion: Overall well written - but if it can be given a better start - it will be good.

7.     Tables and Figures: Tables are good but figures can be improved as they seem stretched out and low in quality.

8.     References and literature review: The review has been fine, but  a lot of important studies have been missed. I would prefer a deeper look and addition of more recent references.

Overall, the article attempts to add important data, but the present version of manuscript cannot be considered for publication unless above concerns are resolved. I hope these comments would help the authors reach a better version of their manuscript.

Comments on the Quality of English Language

The overall quality of writing is good. But there are various instances which can be improved: 

1. In abstract - Replace "multicentre," with "multicentric".

2. Read article again and improve its readablity.

Author Response

We appreciate your review of our paper “A Cross-Sectional, Multicentric, Disease-Specific, Health-Related Quality of Life study in Greek Transfusion Dependent Thalassemia Patients" Your feedback is valuable, and we addressed the issues you raised to enhance the quality and impact of our article.

  1. Novelty and relevance: The article adds important data and is relevant but the authors must enhance the last paragraph of introduction even more - and underline the relevance of study in detail. Make the objectives and endpoints more in line with your title to allow better understanding for the readers. It will also be wise to suggest in a single line how this study is different from already existing body of literature.

We acknowledge your points about the last paragraph in the introduction. We have revised the whole paragraph (lines 96-109) and added the following sentence to underscore the novelty of our study “This research represents the inaugural examination of HRQoL, utilizing a combination of a generic and Thalassemia-specific questionnaire within a representative sample of TDT patients in Greece”

  1. Academic writing, grammar, flow and readability: Overall writing is good. But use of word "multicentre" can better be replaced with "multicentric". Also there are various areas where passive voice has been extensively used. It will be better to report in active voice as that helps better engagement for the reader.

Your comments regarding the phrasing are well-received. We have replaced the term “multicentre” with “multicentric”. We have also made revisions to replace passive voice constructions with active voice throughout the manuscript to enhance readability and clarity for the reader (changes are marked in red color)

  1. Narrative structure: Great structure and headings.

Thank you for your positive feedback

  1. Methodology: This section can be arranged better - the paragraph though descriptive is kind of confusing for the reader. It can better be placed in following headings: Study design and setting, Inclusion and Exclusion criterion, Sampling and Sample Size, Statistical tools and Ethical considerations.

We acknowledge your point about the arrangement of the methods section. We have followed your proposed structure and revised the methods accordingly. Lines 111-199

  1. Statistical tools employed: Good description.

Thank you for your positive feedback

  1. Discussion: Overall well written - but if it can be given a better start - it will be good.

Thank you for your comment. We have revised the first paragraph of the discussion as follows:

“This survey marks a significant milestone as the first comprehensive assessment of the QoL in a representative sample of the adult TDT population in Greece. Employing a combination of the generic SF-36v2 and the Thalassemia-Specific TranQol questionnaires, our study sheds light on the nuanced aspects of HRQoL in this patient cohort………...” Lines 369-381

  1. Tables and Figures: Tables are good but figures can be improved as they seem stretched out and low in quality.

We apologize for the low quality and stretched-out figures. Your detailed instructions are greatly appreciated. We refined the figures as per your suggestions to ensure clear presentation and comprehension.

  1. References and literature review: The review has been fine, but  a lot of important studies have been missed. I would prefer a deeper look and addition of more recent references.

We have added recent references listed below. Initially, we did not include these important studies in our references because the study population included both adults and children. Additionally, following the suggestion of reviewer#2, we have incorporated studies conducted specifically in the pediatric thalassemia population.

  1. Shafie et al. Health and Quality of Life Outcomes (2020) 18:141
  2. Hakeem et al. Health and Quality of Life Outcomes (2018) 16:59
  3. Etemad et al.Hemoglobin 2021 Vol. 45 Issue 4 Pages 245-249
  4. Hossain et al. Scientific Reports 2023 Vol. 13 Issue 1

Overall, the article attempts to add important data, but the present version of manuscript cannot be considered for publication unless above concerns are resolved. I hope these comments would help the authors reach a better version of their manuscript.

Comments on the Quality of English Language

The overall quality of writing is good. But there are various instances which can be improved: 

  1. In abstract - Replace "multicentre," with "multicentric".
  2. Read article again and improve its readablity.

We have thoroughly revised the manuscript to improve readability. All changes are in red.